# A conserved histidine modulates HSPB5 structure to trigger chaperone activity in response to stress-related acidosis

Ponni Rajagopal[1], Eric Tse[2], Andrew J Borst[1], Scott P Delbecq[1], Lei Shi[1], Daniel R Southworth[2]*, Rachel E Klevit[1]*

[1]Department of Biochemistry, University of Washington, Seattle, United States; [2]Department of Biological Chemistry, Life Sciences Institute, University of Michigan, Ann Arbor, United States

**Abstract** Small heat shock proteins (sHSPs) are essential 'holdase' chaperones that form large assemblies and respond dynamically to pH and temperature stresses to protect client proteins from aggregation. While the alpha-crystallin domain (ACD) dimer of sHSPs is the universal building block, how the ACD transmits structural changes in response to stress to promote holdase activity is unknown. We found that the dimer interface of HSPB5 is destabilized over physiological pHs and a conserved histidine (His-104) controls interface stability and oligomer structure in response to acidosis. Destabilization by pH or His-104 mutation shifts the ACD from dimer to monomer but also results in a large expansion of HSPB5 oligomer states. Remarkably, His-104 mutant-destabilized oligomers are efficient holdases that reorganize into structurally distinct client–bound complexes. Our data support a model for sHSP function wherein cell stress triggers small perturbations that alter the ACD building blocks to unleash a cryptic mode of chaperone action.

*For correspondence: dsouth@ umich.edu (DRS); klevit@u. washington.edu (REK)

Competing interests: The authors declare that no competing interests exist.

## Introduction

Cells have numerous strategies to cope with the consequences of stress conditions that lead to protein misfolding and aggregation. Ineffective resolution of protein misfolding and unmitigated protein aggregation can lead to formation of plaques, fibrils, and other aggregated species encountered in neurodegenerative diseases. Generally, ATP-dependent chaperones such as Hsp70 and Hsp90 assist with protein refolding, while ATP-independent chaperones known as small heat shock proteins (sHSPs) act as first responders by maintaining proteins in soluble forms to inhibit misfolding and to delay aggregation. Under transcriptional regulation of the heat shock factor, Hsf1, levels of sHSPs can rise to 1% of total cellular protein when conditions deviate from the norm, e.g. ischemia, hypoxia, oncogene activation, and chemotherapy (*Kampinga and Garrido, 2012*). sHSPs are found throughout prokaryotes and eukaryotes. There are ten sHSPs encoded in the human genome (HSPB1, HSPB2, etc), ranging in size from 15 to 25 kDa. Most, including the ubiquitously expressed human sHSP HSPB5, form large assemblies that exist as dynamic distributions of polydisperse oligomers whose properties are both temperature and pH dependent (*Jehle et al., 2011*). Which form or forms of an sHSP are active and how they function are central unanswered questions. In addition, how sHSPs interact with partly unfolded or aggregate-prone proteins ('clients') to prevent formation of aggregates remains enigmatic.

Some acute cellular stresses are associated with acidosis. For example, measurements in mouse brain following ischemic stroke showed a pH of 6.4 in ischemic tissue compared to a pH of 7.0 in normal tissue (*McVicar et al., 2014*). A decrease in cellular pH may give rise to destabilization of some proteins and/or a decrease in solubility for proteins with pI values between pH 6 and 7. Furthermore,

**eLife digest** Proteins are composed of one or more long chain-like molecules that must fold into complex three-dimensional shapes in order to work properly. Incorrectly folded proteins cannot function and often aggregate into toxic states that are associated with a number of neurological diseases including Alzheimer's, Huntington's, and Parkinson's.

Elevated temperatures, increased acidity, and other stressful conditions in the cell can hinder the folding process and may cause existing proteins to unfold and aggregate. However, when cells experience these stresses, certain proteins—known as small heat shock proteins (or sHSPs for short)—act as 'holdase chaperones' to protect cells from protein misfolding.

HSPB5 is one such chaperone that binds to and stabilizes other proteins (called 'clients') to prevent their aggregation. The core structure of HSPB5 and other similar chaperone proteins is well known. But, it is not clear how chaperones sense stressful conditions and respond to increase their activity to help stabilize client proteins.

Now, Rajagopal et al. have identified a single amino acid in HSPB5 that is sensitive to pH changes. When the environment inside a cell becomes more acidic, this amino acid (a histidine) triggers changes in HSPB5's structure that enhance the chaperone's activity.

This histidine was then replaced with another amino acid in an attempt to lock HSPB5 into a low-pH state that mimics an active HSPB5 chaperone inside a stressed cell. Further experiments revealed that this mutant HSPB5 is a super-active holdase chaperone, and that it dramatically changes its structure to bind to a client protein in the holdase state. From this, Rajagopal et al. propose a model to explain how cellular stress triggers small changes in HSPB5 that propagate through the chaperone in a response mechanism that increases its activity. Future studies will investigate whether inherited mutations in HSPB5 and other similar chaperones—which are associated with cardiac, muscle, and nerve disorders—exert their effect by disrupting this response mechanism.

the pH in normal eye lens fiber cells, which contain extremely high concentrations of the sHSPs HSPB4 and HSPB5 is ∼6.5 (*Bassnett and Duncan, 1985*; *Mathias et al., 1991*). Such observations raise the question as to the nature and determinants of pH-dependent properties of sHSPs in general and HSPB5 in particular.

Here, we report how pH affects structural, biochemical, and functional properties of HSPB5 (also known as αB-crystallin). Like all sHSPs, HSPB5 has a conserved central α-crystallin domain (ACD) flanked by variable N- and C-terminal regions. We carried out investigations both on the HSPB5-ACD, which forms the dimeric building block found in all sHSPs, and on full-length oligomeric HSPB5. We find that ACD dimer stability decreases over a narrow physiologically relevant pH range and we have identified a conserved histidine residue that is largely responsible for the observed destabilization. Surprisingly, the histidine is not on the dimer interface but its mutation to glutamine or lysine partly or fully recapitulates low pH behavior both in the context of the ACD dimer and in oligomers. The mutant proteins allowed us to probe the consequences of dimer interface destabilization without additional complicating factors that arise when comparing experiments performed at differing pH. Mutations that destabilize the dimer interface produce extremely large oligomers that can rearrange to form long-lived complexes with a model client protein. Our studies unmask a cryptic mode of HSPB5/client interaction not previously detected using the wild-type protein under non-stress conditions. The results highlight mechanistic differences in the ways in which holdases work and suggest that HSPB5 has a repertoire of ways in which it can carry out its function.

## Results

### NMR solution structure and $^{15}$N relaxation reveal conformational plasticity in HSPB5-ACD

As ACD dimers are the fundamental, stably folded building blocks of sHSP oligomers, we first sought to define their response to changes in pH over a physiologically relevant range. Although multiple structures have been solved for HSPB5-ACD, there are differences among them as well as differences in

the conditions under which the structures were obtained (i.e., pH 4.6, 6.0, 7.5, 8.5, 9.0) (*Clark et al., 2011*). We felt this situation necessitated structure determination under solution conditions relevant to our studies. The solution structure of HSPB5-ACD at pH 7.5, 22°C (*Figure 1A,B*) was calculated from NMR data that included backbone and side chain chemical shifts, NOEs, and RDCs (*Table 1*). As in all previously determined HSPB5-ACD dimer structures (PDB 2WJ7, 3L1G, 2KLR, 2Y22, 4M5S, and 4M5T), each subunit adopts a six-stranded β-sandwich structure and two ACDs form a dimer through anti-parallel alignment of their long β6+7 strands (*Figure 1A,B,C*). A notable difference among solved sHSP ACD structures is that dimers appear in different alignments of the two β6+7 strands (*Clark et al., 2011*; *Hochberg et al., 2014*). In the solution structure reported here, the dimer interface places Glu117 across from Glu117′ (*Figure 1D*). This register is observed in all but two of the available HSPB5-ACD structures and is also the predominant alignment observed in other metazoan sHSP ACD structures.

At the level of the protomers, the HSPB5-ACD structures are remarkably similar despite having been solved under significantly different conditions and by different techniques (*Figure 1E* and

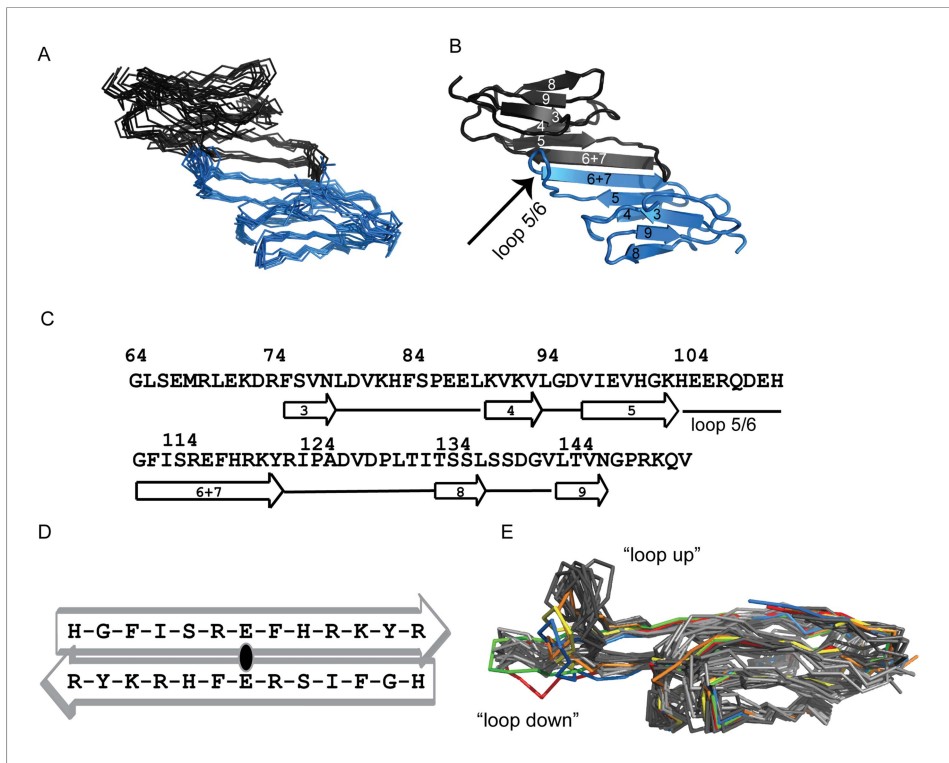

**Figure 1**. Solution structure ensemble of HSPB5-ACD at pH 7.5 is an anti-parallel dimer. (**A**) Backbone traces of ten ACD structures determined by RosettaOligomer are aligned over all residues (RMSD of 1.4 Å). Subunits of the dimer are shown in blue and gray. (**B**) Cartoon representation of one member of the HSPB5-ACD ensemble is shown. The six strands of the β-sandwich structure are labeled using previously defined nomenclature for ACD structures. The dimer is formed by antiparallel arrangement of the β6+7 strands at the interface. Loop 5/6 is highlighted in the blue subunit. (**C**) The sequence of the ACD construct for which the solution structure was solved is shown with elements of secondary structure highlighted. (**D**) The alignment of the β6+7 strands at the dimer interface, shown schematically, places Glu-117 across from Glu-117′. This alignment is called 'APII' in previous reports (*Clark et al., 2011*). (**E**) Overlay of protomers from all available HSPB5-ACD structures. All members of the solution NMR structure (this paper; PDB 2N0K) and the solid-state NMR structure (2KLR; *Jehle et al., 2010*) are shown in dark gray and light gray, respectively. Five crystal structures are shown: (1) 2WJ7 (yellow), (2) 3L1G (orange), (3) 4M5S (green), (4) 4M5T (red), and (5) 2Y22 (blue).

The following source data are available for figure 1:

**Source data 1**. Comparison of the NMR solution structure and published structures.

**Source data 2**. Multiple sequence alignment of the ten human sHSPs.

**Table 1.** NMR data and refinement statistics for HSPB5-ACD structures

| NMR distance and dihedral constraints | HSPB5-ACD |
| --- | --- |
| Distance constraints | |
| Total NOE | 838 |
| Intraresidue | 310 |
| Inter-residue | |
| Sequential ($|i − j| = 1$) | 255 |
| Medium-range ($|i − j| < 4$) | 85 |
| Long-range ($|i − j| > 5$) | 188 |
| Inter-molecular | 36 |
| Total dihedral angle restraints | |
| φ (TALOS) | 72 |
| ψ (TALOS) | 72 |
| Residual Dipolar Couplings (RDCs) | |
| $^1H$-$^{15}N$ RDCs | 81 |
| Structure statistics | |
| Violations (mean ± s.d.) | |
| Distance constraints (Å) | 0.48 ± 0.45 |
| Dihedral angle constraints (°) | 14.4 ± 14.7 |
| Average pairwise r.m.s deviation (Å)* | |
| Heavy | 2.46 ± 0.97 |
| Backbone | 1.48 ± 0.6 |

*Average pairwise r.m.s.d. was calculated among ten refined structures.

Figure 1—source data 1). Two NMR ensembles (solution-state reported here and solid-state PDB 2KLR) and five crystal structures all overlay well with the exception of the loop that connects β5 and β6+7 (called Loop 5/6; residues His-104–Gly-112). Loop 5/6 curves upwards in our solution ensemble (pH 7.5) and in three crystal structures (2WJ7 [pH 9.0], 3L1G [pH 4.6], and 2Y22 [pH 8.6]). In the 'loop up' conformation, Loop 5/6 residues make contacts across the dimer interface to the other subunit. In contrast, Loop 5/6 curves slightly downwards in the solid-state NMR ensemble (2KLR; pH 7.5) and in two crystal structures (4M5S [pH 6.0]; 4M5T [pH 6.5]). Thus, among structures solved between pH 6.5 and 7.5, both loop conformations have been observed, suggesting that the loop is dynamic in the physiological pH range.

To investigate dynamic processes in the HSPB5-ACD, we performed $^{15}N$ relaxation measurements. A majority of backbone amide nitrogens have $^{15}N$ $T_2$ values between 30 and 40 ms, but resonances in Loop 5/6 and the dimer interface have $T_2 < 30$ ms, indicating they undergo a change in their environment in the millisecond timescale (data not shown). To better probe this time regime, we performed NMR $^{15}N$ relaxation-compensated CPMG relaxation dispersion measurements under the solution structure conditions (700 μM ACD at pH 7.5, 22°C). Examples of relaxation curves obtained from spectra collected at two magnetic field strengths (600 and 800 MHz) are shown in **Figure 2A**. Twenty resonances gave curves that could be analyzed; of these, half are well fit by a two-state model ($\chi^2_{red}$; reduced chi-square < 10; **Figure 2—source data 1**). This indicates that the ACD dimer is exchanging between a major species (i.e., the one revealed in the solution structure ensemble) and one or more minor species. The nature of the minor species will be discussed below. Values for $k_{ex}$, the rate of exchange, for individual resonances ranged between 760 s$^{−1}$ and 1119 s$^{−1}$. The less than two-fold difference for the range of values obtained from fitting data for individual resonances suggests that the residues that exhibit exchange could be involved in the same dynamic phenomenon. The residues that undergo exchange are along the dimer interface and in Loop 5/6 (**Figure 2B**). These findings are consistent with the apparent plasticity of the dimer interface and Loop 5/6 inferred from crystal structures of the ACD dimer. Analysis of available HSPB5-ACD structures and NMR relaxation data combined indicate that (1) the HSPB5 protomer structure is retained over a wide pH range and (2) there are two regions of plasticity, namely the dimer interface and Loop 5/6.

## HSPB5-ACD undergoes pH-dependent dimer dissociation

To assess thermodynamic stability of the ACD dimer, dimer–monomer dissociation constants were determined from isothermal titration calorimetry dilution measurements in which highly concentrated ACD was titrated into buffer. The resulting isotherms fit a simple dimer-monomer equilibrium model (**Table 2**). We made measurements at two physiological pH values, pH 7.5 and pH 6.5 (25°C), and obtained dissociation constants of 2 ± 2 μM and 30 ± 16 μM, respectively. The value at pH 7.5 is in excellent agreement with the value of 2 μM inferred from tandem MS/MS measurements performed at the same pH (**Hochberg and Benesch, 2014**). The dimer is also destabilized by an increase in temperature, with a $K_D$ of 36 ± 2 μM at pH 7.5, 37°C. Thus, the HSPB5-ACD dimer is destabilized by either a decrease in pH from 7.5 to 6.5 or an increase in temperature from 25°C to 37°C.

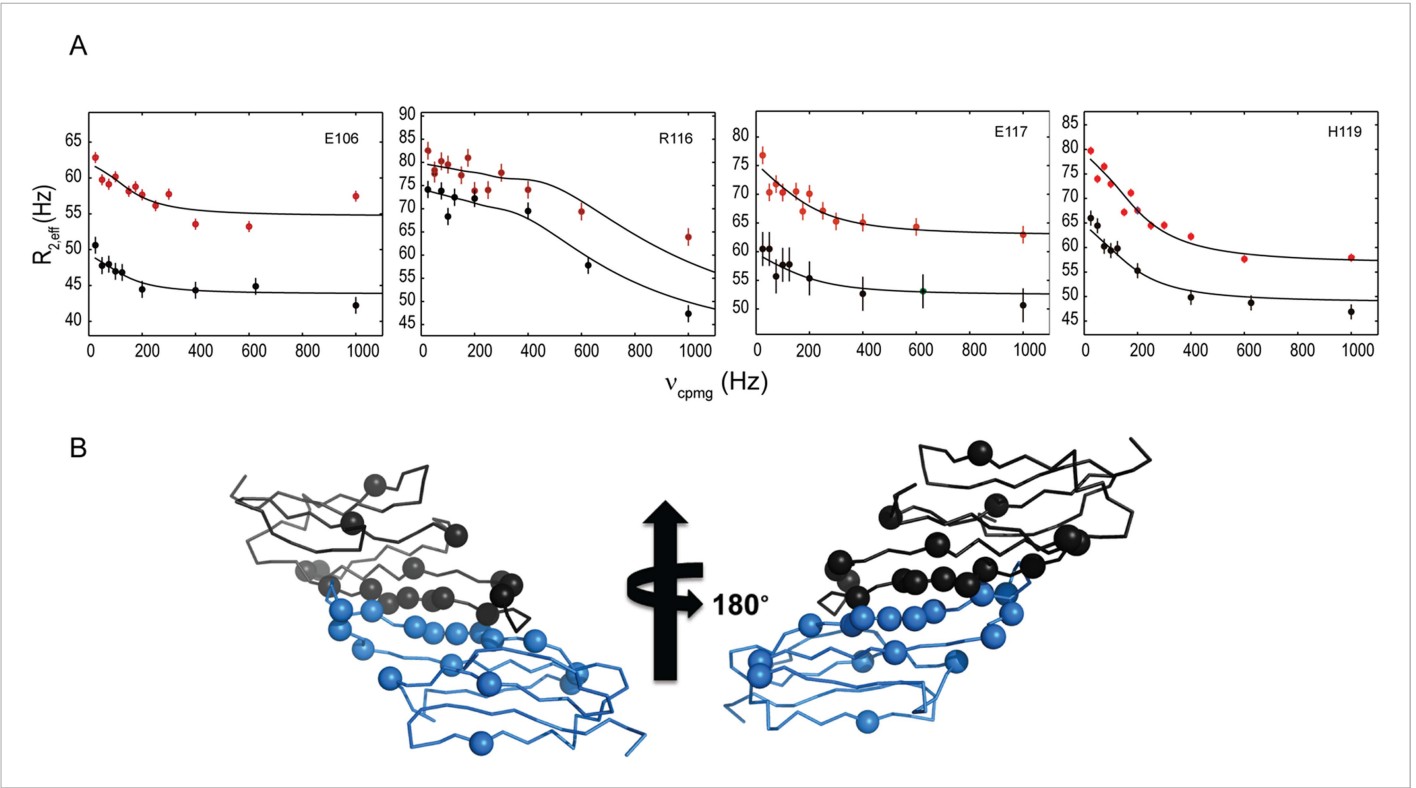

**Figure 2**. $^{15}$N-CPMG relaxation dispersion experiments detect dimer-to-monomer exchange. (**A**) Relaxation dispersion measurements reveal a two-state transition. Representative relaxation dispersion curves of $^{15}$N transverse relaxation rate ($R_{2,eff}$) plotted as a function of field strength, ($\nu_{CPMG}$) are shown (see Materials and methods for details). Data were recorded at static field strengths of 800 MHz (red) and 600 MHz (black) at pH 7.5 and 22°C. Values of $k_{ex}$, $\delta\omega$, and $pb$ were extracted using the program, GUARDD. (**B**) Backbone representation of HSPB5-ACD dimer is shown with the Cα atoms of exchanging residues that are well fit by a two-state model shown as spheres. Resonances showing relaxation rates in the range 760 s$^{-1}$ to 1119 s$^{-1}$ occur mainly at the dimer interface and in Loop 5/6.

The following source data is available for figure 2:

**Source data 1**. Parameters from relaxation dispersion experiments performed on 0.7 mM HSPB5-ACD at 22°C and pH 7.5.

$^1$H, $^{15}$N-HSQC spectra of HSPB5-ACD collected at pH 7.5 and 6.5 show extensive changes (**Figure 3A**). Furthermore, spectra collected between pH 7.0 and pH 6.0 had significant peak doubling which is mostly resolved at pH 6.0, making analysis of the pH-induced spectral changes based solely on existing resonance assignments at pH 7.5 (*Jehle et al., 2009*) challenging. Therefore, we assigned the HSPB5-ACD spectrum at pH 6.5 using standard triple resonance spectra collected at both pH 6.5 and pH 6.0 so that we could follow individual resonances through a pH titration series. An expanded region of the $^1$H, $^{15}$N-HSQC pH series is shown in **Figure 3B**. In the absence of another pH-dependent process, resonances arising from residues that undergo a protonation/deprotonation event as a function of pH will shift in a continuous manner. Resonances from residues that are proximal to titrating residues will also show similar behavior. Such processes will appear in the so-called 'fast-exchange' NMR regime due to the rapid on/off rate of acidic protons. At pH values above 6.7, some resonances in the HSPB5-ACD spectrum exhibit fast-exchange pH behavior, consistent with protonation/deprotonation of ionizable groups. This behavior is illustrated by the resonance for

**Table 2**. Dimer-to-monomer dissociation constants for HSPB5-ACD determined by ITC*

| Temperature | pH | $K_d$ (mM) | ΔH (cal/mol) |
|---|---|---|---|
| 25°C | 7.5 | 0.002 ± 0.002 | 8772 ± 3960 |
| 25°C | 6.5 | 0.030 ± 0.016 | 3242 ± 432 |
| 37°C | 7.5 | 0.036 ± 0.002 | 9328 ± 527 |

*See 'Methods and materials' for experimental details and data analysis.

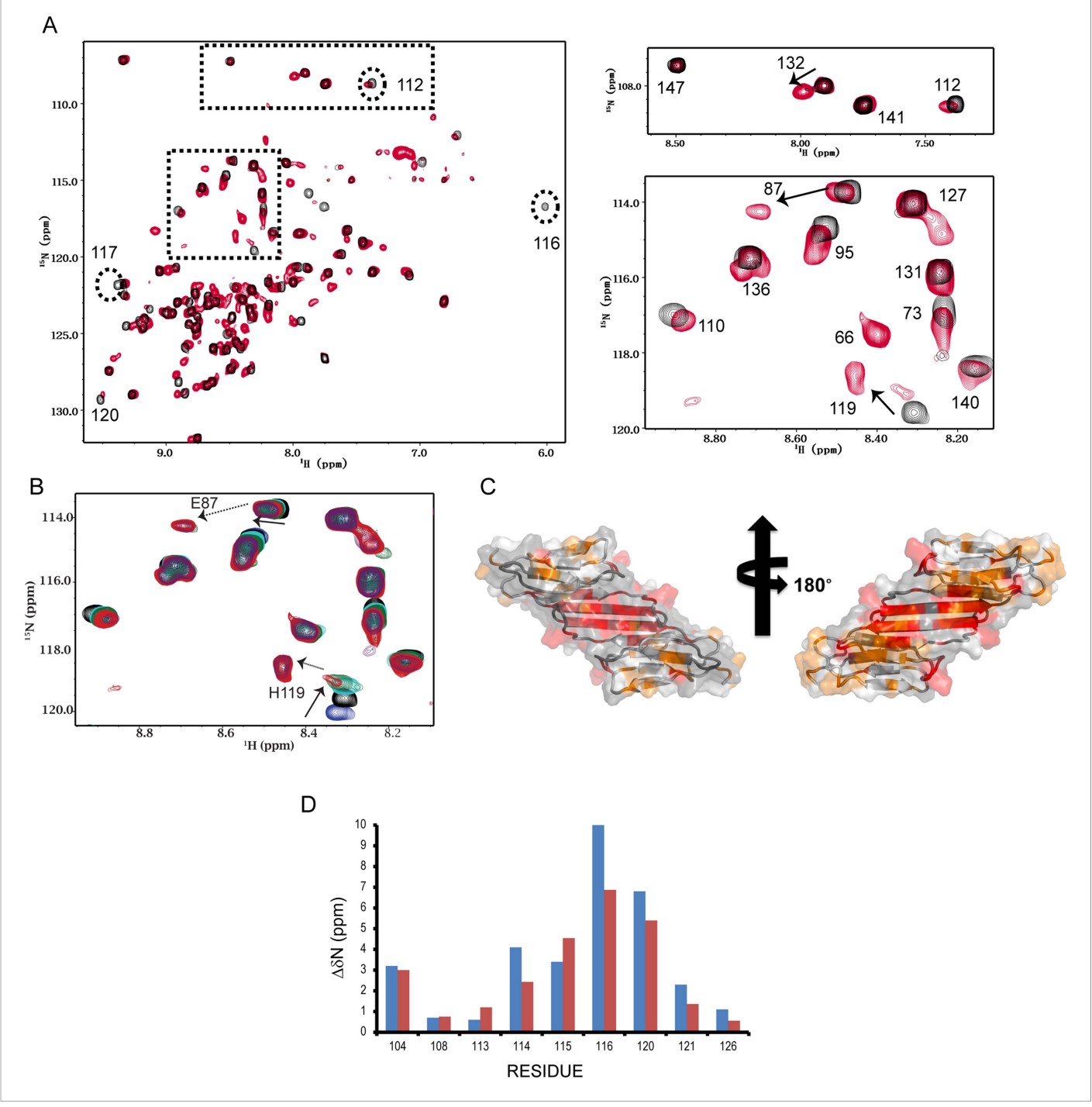

**Figure 3**. HSPB5-ACD undergoes a conformational transition between pH 7.5 and 6.5. (**A**) $^1$H-$^{15}$N HSQC spectra acquired on a 200 μM sample of HSPB5-ACD at pH 7.5 and 6.5 (22°C) reveal two states. Full spectra collected at pH 7.5 (black) and 6.5 (red) are overlaid (left). At pH 7.5, 82/85 non-proline residues are observed (residues G64, L65, and S139 are not detected due to fast exchange with H$_2$O). At pH 6.5, approximately ~65 additional peaks appear, indicating the presence of two conformations. Boxed regions shown on the right provide clear examples of peak doubling. Some resonances (labeled in the panel on the left) disappear from their original positions at pH 7.5 and their new positions could not be determined by inspection of the spectrum at pH 6.5. (**B**) Example of a full pH titration series (spectra collected at pH 8.4 [blue], 7.5 [black], 7.0 [cyan], 6.7 [green], and 6.5 [red]). The behavior of the resonance of H119 as a function of pH illustrates two pH-dependent processes (see text). Its chemical shift is in fast exchange from pH 8.4 to 7.5 (solid arrow) and changes direction and is in slow exchange from pH 7.0 to 6.5 (dashed arrow). The region shown contains several other resonances that undergo slow-exchange transitions over the same pH range. (**C**) Residues that undergo the slow exchange transition are highlighted in color on a surface representation of the ACD dimer. Residues with perturbations > 0.2 ppm are red; those with perturbations between 0.1 - 0.2 ppm are orange. (**D**) Analysis

*Figure 3. continued on next page*

Figure 3. Continued

of relaxation dispersion data yields the difference in $^{15}$N chemical shift between the major and minor species ($\Delta\delta N_{calc}$). The values of $\Delta\delta N_{calc}$ (blue) are compared to the experimental values obtained from the difference of $^{15}$N chemical shifts at pH 7.5 and 6.5, i.e., $\Delta\delta N^{(7.5-6.5)}$ (red), in the histogram. Concordance between these two parameters supports the notion that the minor form detected by relaxation dispersion experiments is the monomeric form of ACD that is populated at lower pH.

His-119 in spectra collected at pH 6.7 and above (*Figure 3B*). At pH < 6.7, some resonances double and appear in a new position (for example, resonances labeled 87, 119, and 135 in *Figure 3B*). Such behavior exemplifies intermediate-to-slow exchange and is indicative of the existence of two states that interconvert slowly on the NMR timescale. The relative intensities of the peaks reflect the relative population of the two states, so a new species of HSPB5-ACD is increasingly populated as the pH decreases.

Spectral assignments at pH 7.5 and 6.5 allowed identification of residues most affected by the pH-dependent transition. The chemical shift difference between the 'high' pH form (pH 7.5) and the 'low' pH form (pH 6.5), $\Delta\delta(pH^{7.5}-pH^{6.5})$ was calculated for each NH resonance (see 'Materials and methods'). Resonances that exhibit large chemical shift perturbations (>0.2 ppm) and/or change trajectory during the titration mainly arise from two structural regions (colored red in *Figure 3C*): the dimer interface (residues F113–I124) and Loop 5/6. In light of the pH-dependent $K_D$ values obtained by ITC, the dissociation of dimers to monomers is a likely source of the slow exchange process observed in NMR spectra below pH 7.

The above findings raised the possibility that the minor species detected by relaxation dispersion at pH 7.5 is an ACD monomer. In addition to the exchange rate, the fractional population of the minor species ($pb = 1 - pa$), and the chemical shift difference between the major and minor species for a given resonance ($\delta\omega$) can be extracted from analysis of relaxation dispersion data (*Kleckner and Foster, 2011*). Seven resonances provided an estimate for the fractional population of the minor species, $pb$, as ~5%, in good agreement with the value of 4% calculated from the $K_D$ measured by ITC. We compared the predicted difference in $^{15}$N chemical shift ($\delta\omega$) to the experimentally determined difference in chemical shifts between the ACD at pH 7.5 and at pH 6.5, $\Delta\delta N(pH^{7.5}- pH^{6.5})$. As seen in *Figure 3D*, there is remarkably close correspondence for dimer interface residues, supporting the notion that the minor species detected by relaxation dispersion at pH 7.5 is the same as the monomeric form populated as the pH decreases below 7.

Finally, we performed relaxation dispersion measurements at 37°C (pH 7.5) to determine the rates of association/dissociation at physiological temperature. For a dimer-to-monomer transition, the measured exchange rate will be concentration dependent:

$$k_{ex} = k_{md}[\text{monomer}] + k_{dm},$$

where $k_{md}$ is the rate constant for monomer–dimer association and $k_{dm}$ is the constant for dimer dissociation. We collected relaxation dispersion data at two protein concentrations (700 and 200 μM; pH 7.5, 37°C). Global fitting of relaxation dispersion data for residues in the dimer interface gave $k_{ex}$ values of ~1500 s$^{-1}$ and 460 s$^{-1}$ at 700 and 200 μM HSPB5-ACD, respectively. The concentration dependence confirms that the observed exchange is due to dimer-monomer dissociation/association. Based on a $K_D$ (=$k_{dm}/k_{md}$) of 36 μM, the concentration of ACD monomers at 700 μM ACD subunits is ~100 μM. This gives $k_{md} = 1.1 \times 10^7$ M$^{-1}$ s$^{-1}$ (in the range for diffusion-controlled bimolecular association; *Northrup and Erickson, 1992*) and $k_{dm} = 400$ s$^{-1}$. Thus, several NMR parameters plus ITC measurements provide a picture in which the long dimer interface dissociates at a rate of 400 s$^{-1}$ at pH 7.5, 37°C and is destabilized as pH decreases below 7. Under the same conditions, full-length HSPB5 subunits in oligomers exchange much more slowly, at a rate of 10$^{-3}$ s$^{-1}$ (*Peschek et al., 2013*). Therefore, the rate at which subunits leave an oligomer is not determined by disruption of the ACD dimer and the results suggest that dimer interfaces within oligomers may break and reform many times during the residency of subunits within HSPB5 oligomers.

## A conserved histidine is responsible for destabilization of the HSPB5-ACD dimer

The arrangement of the ACD dimer creates patches of positively and negatively charged surface that cross the dimer interface on both faces of the structure (*Figure 4A*). At pH 7.5 and 22°C where the

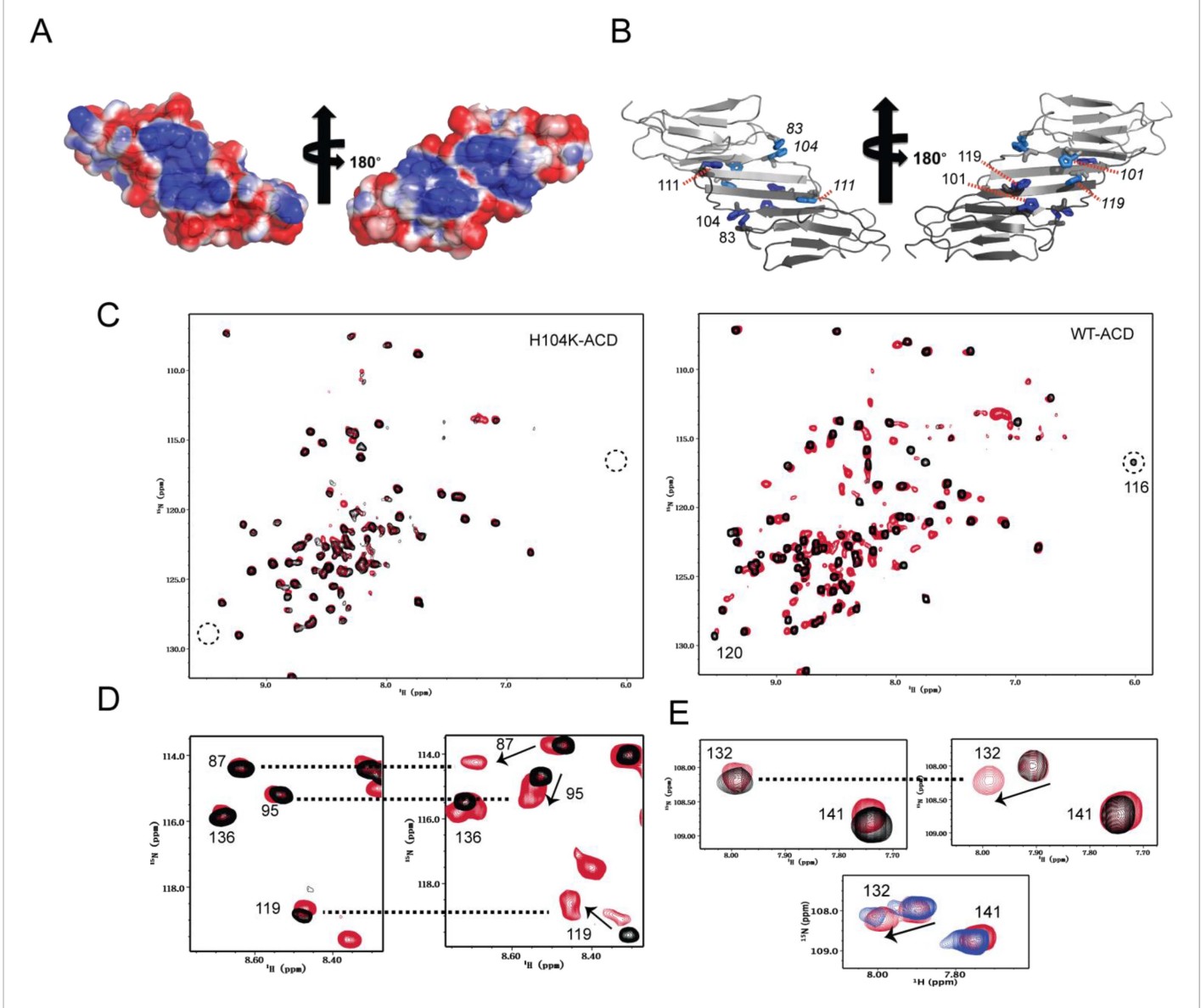

**Figure 4**. His-104 plays a key role in the dimer-monomer transition. (**A**) The electrostatic surface of HSPB5-ACD at pH 7.5 (calculated using experimentally determined histidine $pK_R$ values) reveals patches of positive (blue) and negative (red) charges that cross the dimer interface. (**B**) The five histidines of HSPB5-ACD (blue and cyan sticks) occur as pairs and are located in proximity to the dimer interface. (**C**) Mutation of His-104 shifts the dimer-monomer equilibrium as observed by NMR. Overlays of $^1$H-$^{15}$N HSQC spectra at pH 7.5 (black) and pH 6.5 (red) are shown for H104K-ACD (left panel) and WT-ACD (right panel). A single set of peaks is observed in WT-ACD at pH 7.5, and peaks due to the monomer conformation appear as the pH is lowered. Peaks belonging to the dimer interface, e.g., R120 and R116 disappear from their original positions in WT-ACD at pH 6.5 and in H104K-ACD at both pH conditions (dotted circles). (**D, E**) Expanded regions of $^1$H-$^{15}$N HSQC spectra of WT-, H104K-, and H104Q-ACD. The same color scheme is used for WT- and H104K-ACD as in panel **C**; H104Q-ACD overlay is in blue (pH 7.5) and red (pH 6.5). (**D**) Comparison of H104K-ACD (left panel) and WT-ACD (right panel). Dotted lines connect resonances in similar positions in H104K-ACD and WT-ACD. The example shows that the peaks for residues 87, 95, 119, and 136 have similar positions in H104K-ACD (at both pH values) as in WT-ACD at pH 6.5. (**E**) Comparison of identical regions of spectra of H104K-ACD (top left panel), WT-ACD (top right panel), and H104Q-ACD (lower panel). Both forms (dimer and monomer) are observed in the H104Q-ACD spectrum at both pH values, whereas only peaks corresponding to the monomer are observed in H104K-ACD at both pH values.

The following source data and figure supplement are available for figure 4:

**Source data 1**. The $pK_R$ values and tautomeric states of HSPB5-ACD histidine side-chain imidazole rings (22°C) are listed.

**Figure supplement 1**. His-104 is at the center of a dynamic network of charged and H-bonding interactions.

dimer is more stable, the electrostatic forces are presumably in balance, but the high densities of like charges suggest the stability is tenuous. A protonation event or other rearrangement at or near the dimer interface may tip the balance and trigger dimer dissociation. HSPB5-ACD has a high histidine content, with five His residues out of 89 total residues, (5.6%, which is more than double the average His frequency across all proteomes). In the dimer structure, eight of ten histidine residues (five in each protomer) are located in pairs (two His-83/His-104 pairs and two His-101/His-119 pairs) and are concentrated towards the center of the dimer (*Figure 4B*). Overall, the organization and surface electrostatics suggested a possible role for histidine residues in triggering the pH-dependent dimer dissociation.

To identify candidate histidines, p$K_R$ values and tautomeric states were determined from NMR chemical shifts of imidazole ring nuclei (N$\delta$1, N$\epsilon$2, H$\delta$2, and H$\epsilon$1) measured as a function of pH. p$K_R$ values were obtained for four of five His residues: His-101 and His-111 have low values of <6 and His-83 and His-119 have values of 6.6 and 7.7, respectively (*Figure 4—source data 1*). A p$K_R$ could not be determined for His-104 because its side chain resonances broaden and disappear at pH values below 7.5. Though His-104 is not positioned on the dimer interface, the behavior of its backbone NH resonance is similar to backbone NH resonances on the dimer interface undergoing exchange between dimer and monomer, indicating that it undergoes a change in environment as a result of the dimer-to-monomer transition. At pH 7.5 where its imidazole resonances are detectable, His-104 exists as the less common N$\delta$1H tautomer, while the other four His residues are in the more common N$\epsilon$2H tautomer. A resonance is observed at a chemical shift of 12.4 ppm for the N$\delta$1H of His-104, indicating that it is hydrogen bonded. Among members of the solution ensemble, there are several potential H-bonding partners in proximity to His-104: the backbone carbonyls of His-83, Glu-105, and Glu-106 and side chain groups of Arg-107 and Arg-116 (*Figure 4—figure supplement 1*). Together, the data reveal that His-104 is in a specific conformation in the ACD dimer at pH 7.5 that is stabilized through an H-bond.

The sensitivity of His-104 resonances to the dimer-to-monomer transition suggested it as a prime candidate for the pH trigger. His-104 was substituted with either Gln or Lys and the mutant ACDs were compared to WT-HSPB5-ACD. As discussed above (*Figure 3A,B*), peak doubling is observed in the spectrum of WT ACD below pH 7, with new peaks appearing at lower pH corresponding to ACD monomers. The $^1$H-$^{15}$N HSQC spectrum of H104Q-ACD has peak doubling even at pH 7.5 and the two sets of peaks correspond to dimer and monomer peaks already assigned in the WT spectrum (*Figure 4E*, lower panel). The H104K-ACD HSQC spectrum has a single set of resonances under both pH conditions, but the peaks correspond to the monomer, even at pH 7.5 (*Figure 4C–E*). Thus, substitution of His-104 with glutamine destabilizes the dimer so that approximately equal populations are observed at pH 7.5 and NMR concentrations. Glutamine at position 104 would have difficulty mimicking the N$\delta$1H-bond formed by His-104. Substitution of His-104 with a positively charged lysine yields an ACD that is predominantly monomeric at pH 7.5. We did not determine $K_D$ values for the mutants by dilution ITC experiments because the starting ACD concentration must be ten-fold above $K_D$. Based on the relative populations of dimer and monomer peaks in the NMR spectra, we can estimate that the $K_D$ for H104Q-ACD is ~100–500 μM (approximately equal populations of dimer and monomer at [ACD] = 200 μM) and the $K_D$ for H104K-ACD is >1 mM (based on our inability to detect dimer resonances in samples containing 200 μM protein). Although only order-of-magnitude, these estimates reveal that substitution of His-104 with Gln or Lys destabilizes the dimer more than 100-fold. To ascertain whether the effect of mutating His-104 is specific, the other four His residues were each mutated to Gln. The mutant ACDs retained the pH-dependent spectral behavior of the WT-ACD (data not shown). Altogether, the data implicate His-104, which sits at the base of Loop 5/6 (*Figure 4B*), as a hotspot whose conformation, H-bonding capacity, and protonation state strongly affect the stability of the HSPB5-ACD dimer interface.

## A destabilized dimer interface triggers a large expansion in HSPB5 oligomers

To uncover consequences of dimer interface stability in HSPB5 oligomer structure, we compared properties of full-length H104Q- and H104K-HSPB5 with WT-HSPB5. Size exclusion chromatography combined with multi-angle light scattering (SEC-MALS) provided information on oligomer size and polydispersity and negative-stain electron microscopy (EM) allowed observation of single oligomeric particles (*Figure 5*). SEC-MALS data of WT-HSPB5 oligomers confirmed previously reported

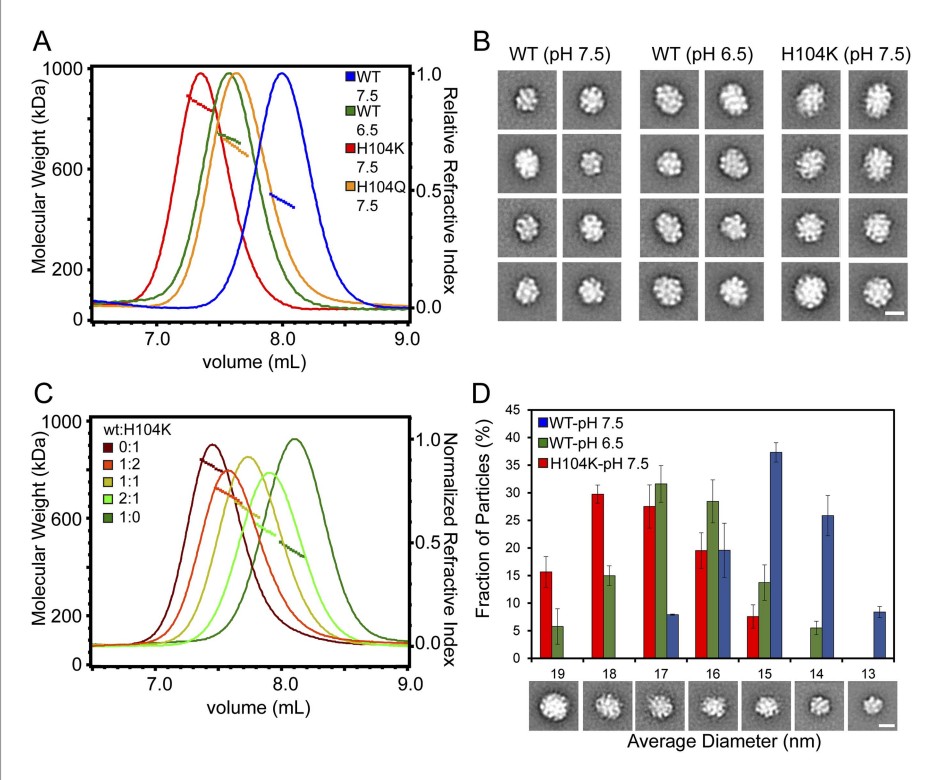

**Figure 5**. Destabilizing the ACD dimer interface via low pH or His-104 mutation triggers large expansion of HSPB5 oligomers. (**A**) SEC-MALS analysis showing protein elution profile (refractive index, right Y-axis) with average $M_w$ (horizontal trace under peak corresponds to left Y-axis) for WT-HSPB5 at pH 7.5 (blue) and pH 6.5 (green), and H104Q-HSPB5 (orange) and H104K-HSPB5 (red) at pH 7.5. (**B**) 2D projection class averages showing representative ensemble of oligomer sizes. (**C**) SEC-MALS analysis showing elution profile and average Mw of mixed oligomers of WT and H104K-HSPB5 incubated together at ratios of 1:0 (green), 2:1 (light green), 1:1 (yellow), 1:2 (orange), and 0:1 (dark red), respectively. (**D**) Histogram of oligomer diameters showing fraction of oligomer particles vs average diameter (nm) for WT at pH 7.5 (blue) and pH 6.5 (green), and H104K at pH 7.5 determined by negative-stain EM 2D classification. Representative 2D class averages corresponding to measured diameter are shown. Experiments were performed in triplicate. Scale bar (lower left image) equals 10 nm.

The following figure supplement is available for figure 5:

**Figure supplement 1**. EM micrograph images, 2D classification, and particle diameter estimation for WT and H104K HSPB5.

pH-dependent changes in oligomeric dimension (*Jehle et al., 2011*; *Baldwin et al., 2011a*). At pH 7.5, the molecular weight ($M_w$) determined at intervals across the elution peak ranged from 440 kDa to 500 kDa. The average $M_w$ of 465 kDa corresponds to the mass of ~24 subunits, consistent with previous reports (*Aquilina et al., 2003*; *Peschek et al., 2009*). At pH 6.5, WT-HSPB5 oligomers elute earlier, with an average $M_w$ of 720 kDa, corresponding to ~36 subunits per oligomer—an increase of 12 subunits on average per oligomer (*Figure 5A*). Negative-stain EM micrographs and single particle images collected on WT-HSPB5 oligomers at pH 7.5 and 6.5 reveal large globular structures at both pHs (*Figure 5—figure supplement 1A*). In 2D projection averages (generated by reference-free alignment of single particle data sets), WT-HSPB5 oligomers are generally spherical and have varied diameters that are on average greater at pH 6.5 than at pH 7.5 (*Figure 5B*, *Figure 5—figure supplement 1B*). Thus, SEC-MALS and EM measurements reveal an expansion in the hydrodynamic radius, particle dimensions, and number of subunits in WT-HSPB5 oligomers as a function of decreasing pH.

To ask if the change in oligomer size observed with pH is a consequence of dimer interface destabilization, the His-104 mutants were analyzed by SEC-MALS and EM at pH 7.5. H104K-HSPB5

forms very large, polydispersed oligomers ranging from 785 to 840 kDa (*Figure 5A*), with an average $M_w$ of 810 kDa (~41 subunits). H104Q-HSPB5 oligomers elute at an intermediate position relative to WT- and H104K-HSPB5. Thus, more subunits are recruited when oligomers are assembled from destabilized dimers. We propose that oligomers composed of H104K-HSPB5 are made predominantly of monomeric units while oligomers composed of H104Q-HSPB5 contain both dimeric and monomeric units. If this model is accurate, we should be able to recapitulate intermediate-sized H104Q-containing oligomers by mixing together WT- and H104K-containing subunits. As shown in *Figure 5C*, the elution position and oligomer mass in mixtures of WT-HSPB5 and H104K-HSPB5 titrate as a function of added H104K-HSPB5 subunits, consistent with a model in which oligomers are formed from both dimeric and monomeric units. WT-HSPB5 oligomers at pH 6.5 have a mass range that corresponds to a 1:2 ratio of WT:H104K subunits, indicating that the pH-dependent oligomeric assemblies likely arise from a combination of monomeric and dimeric subunits.

The results above suggest that H104K-HSPB5 oligomers represent an endpoint in a continuum of oligomer composition and that H104Q- and (pH 6.5) WT-HSPB5 are reflective of species that are favored under differing conditions including stress-induced acidosis. To ask if oligomers composed uniquely or partly of monomeric units have distinct structural features, negative-stain EM images of H104K-HSPB5 at pH 7.5 and WT-HSPB5 at pH 6.5 were collected and analyzed. In a majority of images, defined features, identified as light and dark regions of density, are apparent within the structures, indicating an organized arrangement of subunits. The particles are generally spherical, with some oblong-shaped structures also observed. Quantitative information on the size and distribution of oligomers was obtained by measuring the diameter of circular averages generated for each 2D projection average (*Figure 5—figure supplement 1C*). Single particles were then binned based on diameter of their corresponding average to generate a distribution (*Figure 5D*). The size distribution derived in this way is consistent with the SEC-MALS analysis: WT-HSPB5 at pH 7.5 has the smallest average diameter, 15 nm, (a range of 13–17 nm), H104K-HSPB5 has the largest, with an average of 17 nm, and WT-HSPB5 at pH 6.5 is intermediate, at 16 nm. Oligomers of WT-HSPB5 at pH 6.5 cover the entire range of diameters measured, between 14 and >18 nm, while the distribution of H104K-HSPB5 is skewed to the largest dimensions, with a majority of particles (>70%) greater than 16 nm. For comparison, the diameter of the WT-HSPB5 cryo-EM model (*Braun et al., 2011*) determined similarly from the circular average measures 15 nm, in good agreement with our measurements. Altogether the results reveal that conditions and/or mutations that affect the stability of the antiparallel dimer interface have substantial impact on HSPB5 oligomer size and structure.

## Oligomers with destabilized dimer interfaces are effective holdases that reorganize to bind client proteins

To ascertain the functional and mechanistic consequences of the expanded oligomer structures described above, standard holdase assays in which aggregation of a model client protein is monitored as a function of time in the absence and presence of sHSP were performed. At pH 7.5, H104Q- and H104K-HSPB5 are more effective than WT-HSPB5 at delaying formation of large aggregates by two model clients, αLactalbumin (αLac) destabilized by addition of DTT, and alcohol dehydrogenase (ADH) destabilized by addition of DTT and EDTA (*Figure 6*). The results indicate that enhanced holdase activity is associated with destabilization of the dimer interface.

To understand how destabilization of the dimer interface yields more effective holdases, we compared how HSPB5 and the mutants interact with clients. We took advantage of the long delay in αLac aggregation afforded by both the WT and mutant HSPB5 species to attempt to detect sHSP-client interactions. Mixtures of αLac and one of the HSPB5 species were analyzed by SEC-MALS following addition of DTT and before the onset of aggregation (*Figure 7*). Although WT-HSPB5 maintains αLac in a soluble form during the time frame of the experiment, no interaction between the two proteins was detected by SEC or SDS-PAGE, indicating that its holdase function is achieved through highly transient interactions (*Figure 7A*). In contrast, a mixture of αLac and H104K-HSPB5 elutes with a dramatically different profile. SDS-PAGE analysis across the broad peak that elutes between 7.5 and 9.2 ml shows that it contains both proteins (*Figure 7B*). There is a broad range of molecular weights across this peak, from >550 kDa to ~250 kDa. A similar but less dramatic change in the elution profile was obtained for a mixture of αLac and H104Q-HSPB5 (*Figure 7C*). Thus, long-lived complexes are formed between αLac and the H104 mutant HSPB5s, and these are markedly smaller than the oligomers that are populated in the absence of client protein.

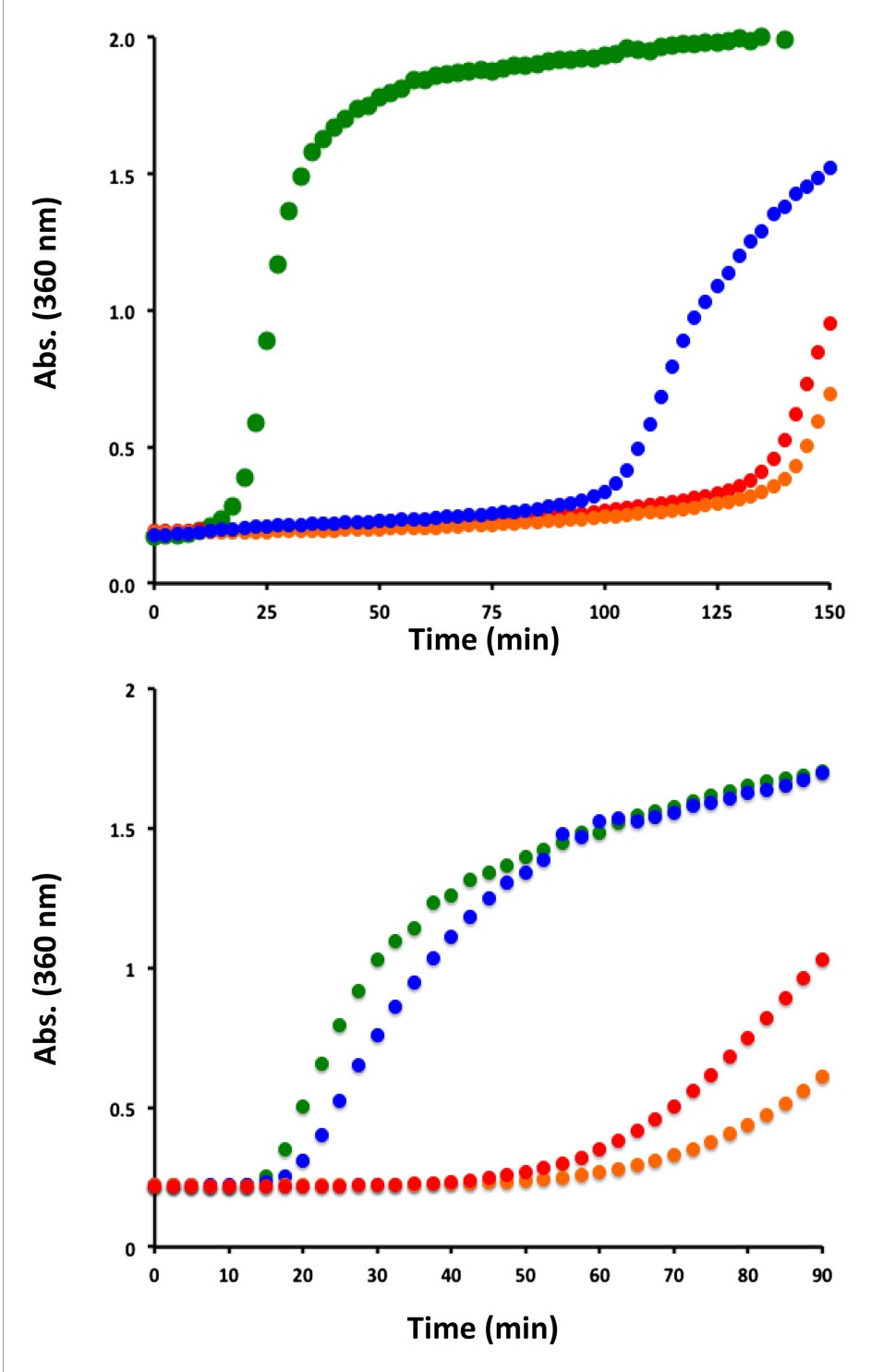

**Figure 6**. His-104 mutants of HSPB5 are effective at delaying the onset of aggregation of a model client protein. (Top panel) Aggregation of DTT-denatured bovine αLactalbumin at 42°C in the absence (green) and presence of WT-HSPB5 (blue) or HSPB5 mutants H104K (red) and H104Q (orange). Light scattering at 360 nm was used to monitor the DTT-induced aggregation of αLac (600 µM) in the presence and absence of HSPB5 (40 µM). Assays were performed in duplicate and the average scattering curves are shown. (Bottom panel) Aggregation of Yeast Alcohol Dehydrogenase in the presence of EDTA and DTT, at 37°C in the absence (green) and presence of WT-HSPB5 (blue) or HSPB5 mutants H104K (red) and H104Q (orange). Light scattering with 360 nm light was used to monitor ADH (100 µM) aggregation in the presence and absence of HSPB5 (20 µM). Assays were performed in duplicate and the average scattering curves are shown.

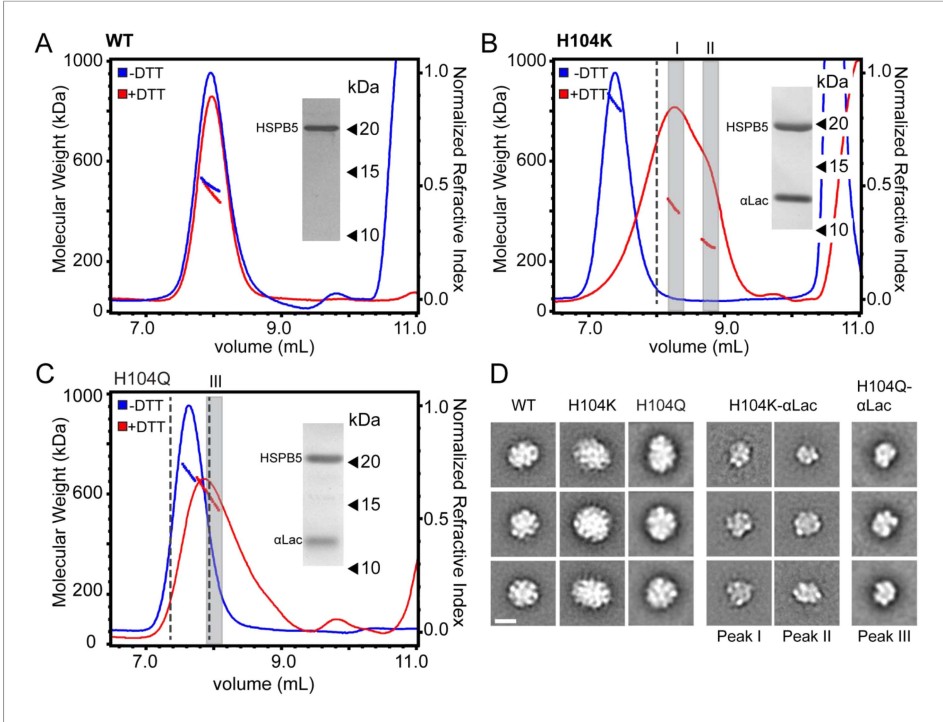

**Figure 7**. H104K- and H104Q-HSPB5 oligomers reorganize into small, long-lived client-bound complexes in the presence of αLac model client protein. (**A–C**) SEC-MALS analysis and corresponding $M_w$ of WT (**A**), H104K (**B**) and H104Q (**C**), HSPB5 oligomers (40 μM in subunit concentration) incubated with αLac (120 μM) in the absence (blue) and presence (red) of 50 mM DTT to destabilize αLac. SDS-PAGE analysis of peak fractions for corresponding HSPB5-αLac incubations with DTT (red trace) is shown (inset). Dashed lines correspond to WT elution at 8 ml and H104K elution at 7.4 ml. (**D**) 2D projection class averages of H104K- and H104Q-HSPB5-αLac peak fractions corresponding to peaks I, II, and III (gray bars) in **B** and **C** are shown with representative averages of WT, H104K-, and H104Q-HSPB5 incubated without αLac substrate for comparison. Scale bar (lower left panel) equals 10 nm.
The following figure supplement is available for figure 7:

**Figure supplement 1**. EM micrograph images and 2D classification of HSPB5 incubated with αLac.

Negative-stain EM was performed on fractions collected across the peak containing both H104K-HSPB5 (or H104Q-HSPB5) and α-Lac (*Figure 7D*). In micrograph and single particle images, the particles appear intact and there is little background from unbound protein, indicating the mutant HSPB5/αLac complexes remain stable following dilution for EM (*Figure 7—figure supplement 1*). The particles are much smaller and are structurally distinct from the spherical oligomers observed for WT-, H104Q-, or H104K-HSPB5 on their own. A large single particle data set of fractionated H104K-HSPB5/αLac complex was collected, and 2D projection averages were determined as above. The 2D averages (*Figure 7D*) show clear structural features and reveal that a major structural reorganization occurs in the mutant sHSP upon binding of a model client protein. Among the 2D averages, some smaller spherical oligomers similar to WT-HSPB5 are observed, but the ultra-large (>16 nm) oligomers that predominate in H104K-HSPB5 alone are no longer detected. The sizes are highly variable, likely depending on the orientation, stoichiometry, and/or oligomeric state of the complex.

To see if similar HSPB5/αLac complexes form with WT-HSPB5 but do not survive the SEC-MALS experiment, unfractionated samples of mixtures of WT-HSPB5 and α-Lac were analyzed by negative stain EM (data not shown). No smaller species were detected and the spherical oligomers remain unchanged for WT-HSPB5. Overall the results demonstrate that HSPB5 oligomers with destabilized dimer interfaces disassemble and undergo a dramatic reorganization to form stable complexes with a model client protein while the wild-type protein at pH 7.5 does not do so and performs its holdase function via highly transient interactions with client.

## Discussion

Despite their key roles in maintaining cellular protein solubility under and following stress conditions, the determinants of sHSP structure and function remain ill defined. Stresses such as ischemia and hypoxia are associated with acidosis in which cellular pH can decrease to as low as pH 6.4 (*McVicar et al., 2014*). As HSPB5 must function under stress conditions, we sought to investigate effects of pH on its structure and function. sHSPs have three structural regions, each of which plays a role in oligomer assembly. The highly conserved ACD is necessary and sufficient for dimer formation and dimers are thought to be the fundamental building blocks for oligomers. The N- and C-terminal regions appear to drive oligomer assembly, although the details of these interactions are not yet elucidated. We find that HSPB5-ACD dimers dissociate into folded monomers that are favored as the pH decreases. ITC and NMR measurements revealed a modest affinity for the ACD dimer ($K_D$ of 36 µM at pH 7.5, 37°C), with dimers dissociating at a rate of 400 s$^{-1}$. The long β6+7 strands of two subunits align in an antiparallel fashion to form the dimer interface. This structure with its associated H-bonds might be expected to afford higher affinity. The juxtaposition of Glu-117 and Glu-117′ in the middle of the interface and the positively and negatively charged patches that cross the interface may contribute to the modest affinity and to the dynamics and plasticity we observed in the interface. Dimer affinity decreases 15-fold over 1 pH unit (pH 7.5 to pH 6.5) and this likely corresponds to a similar fold increase in the rate of dissociation at the dimer interface. A pH-dependent destabilization of the HSPB5-ACD dimer interface was inferred from native MS studies performed on oligomers as a function of pH; our results offer a direct confirmation and quantification of this hypothesis (*Baldwin et al., 2011b*).

We identified a histidine residue that plays a key role in ACD dimer interface stability. Substitution of His-104 with Gln or Lys decreases dimer stability substantially. NMR spectra reveal that the His-104 ring is in the more unusual of the two possible tautomeric forms, stabilized by serving as an H-bond donor. Analysis of members of the solution structure ensemble and crystal structures with the Loop 5/6-up conformation identifies several backbone carbonyls as potential H-bond acceptors for His-104: His-83, Glu-105, and Glu-106. Alteration of the H-bonding potential or geometry (as in the H104Q mutant) or charge (as in the H104K mutant and low pH) destabilizes the dimer. Among the ten human sHSPs, His-104 is the most conserved His residue, appearing in eight of ten proteins, with the non-H104-containing sHSPs having a Gln (HSPB9) or a Lys (HSPB7) (*Figure 1—source data 2*). Consistent with a conserved role for His-104, the unusual tautomeric state adopted by His-104 in HSPB5 is also observed for the analogous histidines in HSPB1 (His-124) and HSPB6 (His-103) (Rajagopal and Klevit, unpublished observation).

Identification of the role of His-104 in dimer stability was unexpected because while there are histidines on the dimer interface, His-104 is not one. His-104 is located at the beginning of Loop 5/6, the loop connecting strands β5 and the dimer interface strand, β6+7 (*Figure 4B*). NMR relaxation dispersion analysis of the ACD dimer revealed that residues in the dimer interface (116–123) exist in two states consistent with a dimer-monomer equilibrium (*Figure 2B*). Residues in Loop 5/6 have similar exchange rates suggesting that loop movements and the structural transition may be coupled. However, a rigorous demonstration that the dynamics of the two regions are coupled requires a robust determination of $k_{ex}$ values as a function of [ACD] for loop residues as well as interface residues and the data collected did not allow for this determination with sufficiently high confidence. Nevertheless, consistent with this notion, inter-subunit contacts are observed that involve Loop 5/6 residues: His-111 at the apex of Loop 5/6 makes contacts to Arg-120′ and Tyr-122′ across the dimer interface. These inter-subunit contacts help to position Loop 5/6 in a 'loop up' conformation in the solution structure at pH 7.5 (*Figure 1E*, *Figure 4—figure supplement 1*). A majority of residues in Loop 5/6 are charged (H$^{104}$-E-E-R-Q-D-E-H$^{111}$) so a possible consequence of His-104 protonation, which sits at the base of the loop, is to alter the conformation or position of Loop 5/6, disrupting inter-subunit contacts involving the loop and shifting the dimer–monomer equilibrium of the HSPB5-ACD. In structures with the 'loop up' conformation, side chains of Arg-107 and Arg-116 are in proximity to His-104; the 'loop down' conformation moves Arg-107 away from His-104 and Glu-106 closer to His-104 (*Figure 4—figure supplement 1*). In sum, we propose that in both loop conformations His-104 sits at the center of a dynamic network of electrostatic and H-bonding interactions that is responsible for modulating the stability of the dimer interface.

A survey of all 17 available mammalian sHSP-ACD structures (9 HSPB5 [WT and mutants], 3 HSPB1, and 3 HSPB6) reveals that Loop 5/6 is found in two conformations; a 'loop up' conformation as seen in

the HSPB5-ACD solution structure and a 'loop down' conformation. In the 'loop up' structures, the details of the interactions between Loop 5/6 residues across the dimer interface vary, again pointing to the plasticity of this region. Nevertheless, the residues corresponding to His-111, Asp-109, and Arg-120 are involved in all cases. Furthermore, although Loop 5/6 contains predominantly polar and/ or charged residues, it is remarkably well conserved among the human sHSPs (*Figure 1—source data 2*). There are several reported examples where Loop 5/6 residues affect the dimer-monomer equilibrium. In HSPB5, residues His-104, Asp-109, and His-111 have recently been implicated in $Cu^{2+}$ and $Zn^{2+}$ binding and the binding of divalent cation appears to destabilize the dimer interface (*Mainz et al., 2012*). In HSPB1, which has an almost identical Loop 5/6 sequence to HSPB5, substitution of the two glutamate residues that correspond to HSPB5 Glu-105 and Glu-106 to Ala resulted in monomeric HSPB1-ACD (*Baranova et al., 2011*). Also, a Loop 5/6 mutation in HSPB1, R127W (corresponding to Arg107 in HSPB5 numbering) is reported to promote monomer over dimer (*Almeida-Souza et al., 2010*). In each case, a change in the net charge of Loop 5/6 leads to monomer being more favored. Altogether the observations suggest a model in which properties of Loop 5/6 are coupled to the stability of the dimer interface. It is notable that inheritance of single alleles of D109H-HSPB5 or R127W-HSPB1 is associated with cataract, myofibrillar myopathy, and distal hereditary motor neuropathy (*Nefedova et al., 2013*). Both mutations alter the net charge of Loop 5/6, implying a critical role for structural and/or dynamic properties of Loop 5/6 in sHSP function.

Moving into the context of full-length HSPB5, we found that decreasing pH or substituting His-104 has dramatic effects on oligomer dimensions and subunit stoichiometry. A continuum of oligomer sizes is observed that trends reciprocally with ACD dimer interface stability. H104K-HSPB5, whose ACD will be predominantly monomeric, assembles into the largest oligomers containing over 40 subunits. While the notion that dimers are the fundamental building blocks of sHSP oligomers may describe the situation under certain conditions, our results indicate that oligomers can also be built from monomeric subunits and from combinations of dimeric and monomeric units and that more monomeric subunits can be incorporated into a given oligomer. This revelation will be important in any future attempts to determine HSPB5 oligomer structures for oligomers larger than the 24-mer previously determined (*Braun et al., 2011*; *Jehle et al., 2011*).

The His-104 mutants allowed us to ask if and how the dimer interface contributes to holdase function without the confounding complications of comparing aggregation of model denatured client proteins at differing pHs. Both dimer-destabilizing mutations yield an HSPB5 that can delay aggregation of model clients longer than the wild-type protein at pH 7.5. Unexpectedly, the His-104 mutant proteins, which form much larger oligomers than WT-HSPB5, reorganize to form long-lived complexes with a client protein. EM images reveal particles that are markedly smaller and distinct from the mostly spherical sHSP-alone structures. During the time period in which WT-HSPB5 inhibits client protein aggregation, we were unable to detect a complex with the client, either by SEC or by EM. Thus, the mutant sHSPB5 species perform holdase function via a different mechanism from the wild-type protein at pH 7.5. WT-HSPB5 acts via weak-and-transient interactions while the mutants act via stronger-and-longer interactions. Notably, two inherited missense mutations in HSPB1 associated with Charcot-Marie-Tooth syndrome, R127W-HPSB1 (in Loop 5/6) and S135F-HSPB1 (on the dimer interface), engage in stronger, longer-lived interactions with client proteins than the WT-HSPB1, as evidenced from tandem affinity purification (*Almeida-Souza et al., 2010*). Two client binding modes have been observed in studies of the highly related HSPB4 (αA-crystallin), dubbed 'high capacity' and 'low capacity' based on differing client:sHSP stoichiometries (*McHaourab et al., 2002*). It remains for future studies to ascertain whether these reflect the species observed and reported here.

Solid-state NMR studies of WT-HSPB5 oligomers at pH 7.5 revealed three types of inter-subunit interactions: (1) ACD-to-ACD (i.e., the dimer interface), (2) C-terminal region-to-ACD, and (3) N-terminal region to either ACD or other N-terminal regions (*Jehle et al., 2011*). The findings in the current study indicate that the relative strength and, perhaps, abundance of these interactions can affect not only the size and structure of oligomers but also the way in which client proteins are recognized. Subunit exchange in WT-HSPB5 is orders of magnitude slower than the exchange rate we measured for the dimer-to-monomer transition ($10^{-3}$ $s^{-1}$ vs ca. $10^3$ $s^{-1}$, respectively, at pH 7.5, 37°C; *Peschek et al., 2013*). So, while breaking the dimer interface is not the rate-limiting step in subunit exchange, destabilizing the ACD dimer yields oligomers that can more readily disassemble to form smaller species with clients and these complexes must be more stable than the mutant sHSP oligomers themselves. Long-lived sHSP-client complexes have been detected by SEC for Hsp18.1

from peas, but the complexes formed are larger than those of Hsp18.1 in the absence of client (*Lee et al., 1997*; *Stengel et al., 2010*). Hsp26 from yeast also forms larger complexes with client than in the absence (*Franzmann et al., 2005*). These examples contrast with our findings, which imply that HSPB5 has evolved to perform its holdase function via weak-and-transient interactions and that small perturbations can unleash a second, cryptic mode of client interaction (long-and-strong). The highly charged ACD dimer interface and the proximal Loop 5/6 provide exquisite sensitivity to small changes in electrostatic environment that can be effected by slight changes in pH, temperature, divalent cations, other cellular conditions, and mutations. Of the approximately twenty inherited disease-related missense mutations documented in ACD regions of human sHSPs, more than half are in residues on the dimer interface or in Loop 5/6 and all these involve substitutions of charged residues.

Our original intent was to define how HSPB5 functions under pH conditions associated with stress-induced acidosis. Our results show unequivocally that the dimer interface stability decreases over a physiologically relevant pH range and that mutation (or protonation) of a single histidine residue is sufficient to destabilize the dimer. The His-104 mutants provide several general insights regarding the modulation of structure and function of sHSPs. First, the stability of an ACD dimer interface relative to other interactions involved in oligomer formation can modulate holdase function. In our study, destabilization of the dimer interface leads to enhanced holdase activity. It remains to be seen if the converse will be true. Second, dimer stability can be modulated by residues other than the dimer interface itself. Such effects are likely achieved through a network of conserved charged residues that ultimately favor or disfavor the dimer over monomer. Third, relatively small changes in pH or single missense mutations are capable of shifting HSPB5 from a weak, transient mode of client interaction to one that involves long-lived co-complexes. We propose that the continuum of oligomeric structures we observe for HSPB5 under differing pH (or mutation) may lead to a continuum of client-binding modes that allow the sHSP to ramp its holdase activity up or down as conditions require. A thorough understanding of the ways in which sHSP structure and activity can be modulated under differing cellular conditions will ultimately provide much needed insights into the cellular functions of this important, but previously intractable class of protein chaperones.

## Materials and methods

### Protein purification and Cloning

HSPB5 and HSPB5-ACD (residues 64-152) were expressed and purified as described previously (*Jehle et al., 2009*). Site-directed mutagenesis was performed with Quik-Change mutagenesis kit from Sigma. The growth and purification protocols of mutant proteins were similar to that of wild-type proteins.

### Structure determination of WT-ACD by solution state NMR

Resonances in the NMR spectrum of WT-ACD were previously assigned (*Jehle et al., 2009*). To obtain distance restraints from NOES, $^{15}$N-edited NOESY and $^{13}$C-edited NOESY spectra on aliphatic and aromatic groups were acquired on a 1 mM, $^{13}$C, $^{15}$N-WT-ACD sample in NMR buffer (50 mM sodium phosphate, pH 7.5100 mM NaCl, 0.1 mM EDTA, and 1 mM PMSF). The spectra were acquired on a Bruker 950 MHz US2 (ultra-shield, ultra-stabilized) spectrometer equipped with Avance III console and a z-gradient, triple resonance cryoprobe (David H Murdock Research Institute in Kannapolis, North Carolina). $^{15}$N-edited NOESY and $^{13}$C-edited spectra were acquired with a mixing time of 120 ms in 90%$H_2O$/10%$D_2O$ solution at 22°C and 100% $D_2O$ solution at 37°C, respectively. Data were processed with NMRPipe (*Delaglio et al., 1995*) and analyzed with NMRViewJ (*Johnson, 2004*) and CcpNmr (*Vranken et al., 2005*). NOEs were binned into short (3 Å), medium (4 Å), and long-range constraints (5 Å) and input as distance restraints into RosettaOligomer (*Sgourakis et al., 2011*). Intermolecular NOEs in homodimeric proteins are usually obtained from edited/filtered-type NOESY experiments on a mixed sample containing labeled and unlabeled protein. This method failed in the case of WT-ACD due to signal-to-noise issues. A preliminary structure of the dimer was determined with CS-Rosetta (*Vernon et al., 2013*) and RosettaDock (*Schueler-Furman et al., 2005*) (see below). The β-sandwich fold was determined from CS-Rosetta using backbone chemical shifts. RosettaDock gave a model of the dimer in the APIII register (where residue R116 from each subunit is across from each other). Using this preliminary model, intra- and inter-molecular NOEs could be parsed out from $^{15}$N-edited and $^{13}$C-edited NOESY spectra. The Hα-Hα NOEs observed in the $^{13}$C-edited NOESY

spectra unambiguously confirmed the APII register as the dimer interface (Glu-117-Glu-117′ across from each other). Intra- and inter-molecular NOE restraints, $^1H$-$^{15}N$ RDCs, and all chemical shifts including the backbone and side-chain were input into RosettaOligomer for the final determination of the WT-ACD dimer structure.

$^1H$-$^{15}N$ residual dipolar couplings (RDCs) were measured on a 500-µM protein sample dissolved in 500 µl of NMR buffer containing 10% Pf1 phage obtained from ASLA biotech. IPAP (In-Phase/Anti-Phase) $^1H$-$^{15}N$ HSQC spectra were acquired in-house on a Bruker Avance III 800 MHz spectrometer equipped with a z-gradient, triple resonance cryoprobe. Spectra were analyzed in NMRView to obtain the values of RDCs. The program PALES (*Zweckstetter et al., 2004*) was used to calculate RDCs for the different structures published in literature.

## RosettaOligomer

The Rosetta symmetric fold-and-dock protocol can be used to determine the structure of symmetric homodimers. Starting from an extended chain, this protocol simultaneously explores the folding and docking degrees of freedom. It consists of four low-resolution stages of increasing complexity in the energy function, in which symmetric fragment insertions are interleaved with symmetric rigid-body trials. During the low-resolution step, side chains are represented using a single, residue-specific pseudo-atom, positioned at the Cα carbon. Finally, symmetric repacking of the side chains and gradient-based minimization of the side chain, rigid body, and backbone degrees of freedom are applied. In this high-resolution step, Rosetta's full-atom energy function is used. The conformational search is largely guided by experimental data, including intra- and inter-molecular NOE distance constraints and RDCs. A penalty term that is proportional to the rmsd between experimental and calculated data was used in Rosetta during the Monte Carlo trials and gradient-based minimization. NOE distances are modeled as atom pair constraints in Rosetta. For a structural model, RDCs are fitted using Levenberg–Marquardt non-linear square fitting algorithm. The orientations of alignment tensor are optimized, while keeping the axial component (Dα) and Rhombic component (R) of the alignment tensor fixed. The values of Dα and R of 21.5 and 0.35, respectively, were estimated from a powder pattern distribution of the RDC data. A total of 10,000 models are generated, of which 1000 lowest ones are selected for cluster analysis. Ten models with the lowest Rosetta full-atom energy in the best-ranked cluster were selected as the final structural ensemble and deposited in the Protein Databank as PDB 2N0K.

## pH titration and determination of pK$_R$ by NMR

$^1H$-$^{15}N$ HSQC-TROSY spectra were acquired on WT-ACD (200 µM) in NMR buffer at pH values 9.3, 8.56, 7.86, 7.44, 7.0, 6.83, 6.7, 6.52, and 6.0 at 22°C on an in-house Avance III 500 MHz spectrometer equipped with a z-gradient, triple resonance probe. The pH of the solution was adjusted by adding small aliquots of 1N HCl or NaOH. The long-range correlations between the ring carbon-bound hydrogens (H$^{\varepsilon 1}$ and H$^{\delta 2}$) and the ring nitrogens (N$^{\varepsilon 2}$ and N$^{\delta 1}$) give information on the tautomeric states of the histidines (*Pelton et al., 1993*). These correlations were observed in $^1H$-$^{15}N$ HSQC spectra using WATERGATE for water suppression and by setting the INEPT delay to an integral multiple of $1/J_{NH}$ where $J_{NH}$ is the value of the single bond N-H coupling constant. pK$_R$ values of histidine residues were determined by following the chemical shifts of H$^{\varepsilon 1}$, N$^{\varepsilon 2}$, and N$^{\delta 1}$ atoms as a function of pH. Non-linear regression fitting of chemical shifts vs pH was performed with Prism (GraphPad) using a modified version of the Henderson–Haselbach equation:

$$\delta_{obs} = \frac{\delta_{HA} + \delta_A - 10^{pH - pK_r}}{1 + 10^{pH - pK_r}},$$

$\delta_{obs}$ is the observed chemical shift at a specific pH value, and $\delta_{HA}$ and $\delta_A$ are the chemical shifts in the fully protonated and deprotonated state, respectively.

## Chemical shift perturbations (CSP) due to pH

CSPs ($\Delta\delta(pH^{7.5} - pH^{6.5})$) were computed as follows:

$$\Delta\delta = 1/2\left(\sqrt{\left((\Delta HN)^2 + (\Delta N/5)^2\right)}\right),$$

where $\Delta\delta$ is the chemical shift difference of an amide group at pH 7.5 and 6.5, $\Delta HN$ and $\Delta N$ are the amide proton and $^{15}$N backbone amide chemical shift differences, respectively.

## $^{15}$N-CPMG relaxation dispersion experiments

To probe conformational fluctuations in the ms timescale, $^{15}$N effective relaxation rates ($R_{2,eff}$ which is the sum of the intrinsic relaxation rate, $R_2^0$ and the chemical exchange rate, $R_{ex}$) were measured for WT-ACD using pulse sequences described in literature (*Korzhnev et al., 2004*). Experiments were performed on 200 and 700 µM samples at 22°C and 37°C at two field strengths, 800 and 600 MHz. The values of $\nu_{cpmg}$ used were 25, 50, 75, 100, 150, 175, 200, 300, 600, and 1000 Hz. A recycle delay of 2.2 s between scans and a total CPMG delay (T) of 20 ms was used. Spectra were processed and analyzed with NMRPipe, and the fits of intensities vs $\nu_{cpmg}$ were analyzed with the program, GUARDD (*Kleckner and Foster, 2012*). 32 out of 85 observable residues in WT-ACD exhibit values of $R_{2,eff}(\infty)$–$R_{2,eff}(0) > 8$ s$^{-1}$ at 22°C, where $R_{2,eff}(\infty)$ and $R_{2,eff}(0)$ are the effective relaxation rates at $\nu_{cpmg}$ values of 25 Hz and 1000 Hz, respectively. Of those, 16 residues could be fit to a two-site exchange model with a reduced $\chi^2 < 10$ and these exhibit two exchange regimes (a) fast exchange (model-2, $k_{ex} >> \delta\omega$) and (b) slow exchange (model-3, $k_{ex} << \delta\omega$). $k_{ex}$ and $\delta\omega$ are the exchange rate and the chemical shift difference between the major and minor state, respectively. In the slow exchange regime, the parameters, $pb$ (the population of the minor state) and $\delta\omega$ can be extracted from the fits in addition to $k_{ex}$. In the fast exchange regime, only the value of $k_{ex}$ can be extracted from the fits. For estimation of goodness of fits, the target function, $\chi^2$ is determined as follows:

$$\chi^2 = \sum_{all\,\nu_{cpmg}} \left(\frac{R_{2,eff}^{obs}(\nu_{cpmg}) - R_{2,eff}^{Calc}(\nu_{cpmg})}{\sigma\left(R_{2,eff}^{obs}(\nu_{cpmg})\right)}\right)^2,$$

where $R_{2,eff}^{Calc}$ is the calculated value of the effective relaxation rate, and σ is the experimental uncertainty in observed $R_{2,eff}$ which is estimated from two data sets that are repeat values of $\nu_{cpmg}$. In this case, repeat data sets were collected at 25 and 200 Hz. The errors reported in $k_{ex}$, $pb$, and $\delta\omega$ values are estimated from 100 Monte Carlo simulations and are reported as the standard deviation of the optimized fit parameter from its 100-element distribution.

## Determination of K$_D$

The value of the dimer–monomer dissociation equilibrium constant, $K_D$, was determined with isothermal titration calorimetry (ITC). ITC was performed on a MicroCal iTC$_{200TM}$ Calorimeter at the Analytical Biopharmacy Core, Univ of Washington. WT-ACD samples were dialyzed into 50 mM sodium phosphate, 100 mM NaCl at either pH 7.5 or pH 6.5. 200 µl of 1 mM WT-ACD and 300 µl buffer were placed in the sample and reference cells, respectively. Forty 1 µl injections were performed and the exothermic heat was measured. Data were fitted using MicroCal Origin software to obtain values of $K_D$ and change in enthalpic heat (ΔH).

## Holdase assays

All holdase assays were performed in duplicate using a 96-well plate reader (BioTek) with PBS solutions at pH 7.5 and 250 µL well volumes. DTT-denatured bovine αLactalbumin (Sigma L6010) was used as a model substrate and light scattering at 360 nm was used to monitor protein aggregation in the presence and absence of WT- and mutant HSPB5 at 42°C. 40 µM sHSP (subunit concentration) was added to 600 µM αLac. Aggregation of αLac was induced by the addition of DTT to final concentrations of 50 mM. Aggregation of the model substrate Yeast Alcohol Dehydrogenase (Sigma A8656) was achieved with the addition of EDTA and DTT to final concentrations of 5 mM at 37°C. Light scattering by aggregates was monitored in the presence and absence of WT- and mutant HSPB5. 20 µM sHSP (subunit concentration) was added to 100 µM Alcohol Dehydrogenase.

## Molecular weight (M_w) determination by SEC-MALS

The $M_w$ of HSPB5 oligomers was determined by separation using a WTC-050S5 SEC column (Wyatt Technology Corporation) with an Akta micro (GE Healthcare) and analysis with a DAWN HELEOS II MALS detector equipped with a WyattQELS DLS, and Optilab rEX differential refractive index detector using ASTRA VI software (Wyatt Technology Corporation). The $M_w$ was determined from the Raleigh ratio calculated by measuring the static light scattering and corresponding protein concentration of a selected peak. Bovine serum albumin served as a calibration standard. Prior to SEC-MALS HSPB5 samples were pre-incubated at 37°C for 30 min at 40 µM monomer concentration in 50 mM sodium phosphate, pH 7.5, 100 mM NaCl, and 1 mM DTT. To examine HSPB5 behavior at pH 6.5, 40 µM of protein was allowed to equilibrate in phosphate buffer (50 mM sodium phosphate, pH 6.5, 100 mM NaCl, and 1 mM DTT) at 37°C for at least 60 min. Substrate binding assays were performed under similar conditions established from αLac holdase assays: HSPB5 (40 µM) was pre-incubated with αLac (240 µM) at 37°C in pH 7.5 buffer for 10 min. DTT was subsequently added to a final concentration of 50 mM to trigger αLac aggregation. After 50 min, the sample was filtered and injected for SEC-MALS analysis and fractionation. Light scattering data and calculations were performed using the ASTRA software package (Wyatt Technology Corporation).

## Negative-stain EM analysis

HSPB5 and HSPB5-αLac samples were diluted to approximately 100–200 nM, applied to a thin carbon-coated copper grid and negatively stained using uranyl formate, pH ∼6.0, essentially as described (*Ohi et al., 2004*). Micrograph images were collected on a Tecnai T12 transmission electron microscope (FEI) equipped with a LaB_6 filament operated at 120 kV. Images were recorded at 50,000X magnification with 2.2 Å/pixel spacing and a 1.0–1.5 µM defocus on a 4k × 4k CCD camera (Gatan). Micrograph images were phase-corrected following CTF estimation and HSPB5 particle projections were selected and excised from micrographs using *EMAN2* (*Tang et al., 2007*). 2D reference-free alignment and classification was performed using *SPIDER* (*Frank et al., 1996*) to generate projection averages. For size estimation, rotational averages were generated from class averages using *SPIDER*. The contrast was normalized for all images and the diameter was measured across the rotational average. The averages were grouped according to size, and the total number of single particles for each group was used to obtain the size distribution. Independent data collection and analysis were performed in triplicate to obtain error bars.

## Acknowledgements

We thank Peter Brzovic, Vinayak Vittal, and Lisa Tuttle for informative discussions and critical evaluation of the manuscript. We thank Katja K Dove and James Fields for collecting biochemical data in support of this project and Kevin Knagge at the David H Murdock Foundation for collecting NMR spectra at 950 MHz. We thank David Baker for access to RosettaOligomer software and computing facilities. The work was supported by NIH grant 1R01 EY017370 (to REK). SD is supported in part by NIGMS 2T32 GM008268.

## Additional information

### Funding

| Funder | Grant reference | Author |
| --- | --- | --- |
| National Institutes of Health (NIH) | R01 EY017370 | Rachel E Klevit |
| National Institutes of Health (NIH) | T32 GM008268 | Scott P Delbecq, Rachel E Klevit |

The funder had no role in study design, data collection and interpretation, or the decision to submit the work for publication.

### Author contributions

PR, Conception and design, Acquisition of data, Analysis and interpretation of data, Drafting or revising the article; ET, SPD, LS, Acquisition of data, Analysis and interpretation of data, Drafting or

revising the article; AJB, Acquisition of data, Analysis and interpretation of data; DRS, REK, Conception and design, Analysis and interpretation of data, Drafting or revising the article

## Additional files

### Major dataset

The following dataset was generated:

| Author(s) | Year | Dataset title | Dataset ID and/or URL | Database, license, and accessibility information |
|---|---|---|---|---|
| Rajagopal P, Shi L, Baker D, Klevit RE | 2015 | Chemical shift assignments and structure of the alpha-crystallin domain from human, HSPB5 | http://www.rcsb.org/pdb/search/structidSearch.do?structureId=2n0k | Publicly available at RCSB Protein Data Bank (2N0K). |

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
