## [Decision Letter]

Thank you for sending your work entitled “A conserved histidine modulates HSPB5 structure to trigger chaperone activity in response to stress-related acidosis” for consideration at *eLife*. Your article has been favorably evaluated by John Kuriyan (Senior editor) and two reviewers, one of whom, Volker Dötsch, is a member of our Board of Reviewing Editors.

The Reviewing editor and the other reviewer discussed their comments before we reached this decision, and the Reviewing editor has assembled the following comments to help you prepare a revised submission.

This is a well written paper reporting, unexpectedly, that a human sHsp highly conserved histidine some distance from the alpha-crystallin domain dimer interface, modulates the stability of that interface as a function of pH, over a physiological range. The authors provide a range of convincing evidence, and also show that the effect is specific to His-104 (in the sequence of αB-crystallin). They provide a well argued overview of likely correlated structural changes. Using single site mutations they provide evidence that loss of His-104 results in large oliogmers built predominantly from monomers that function as stronger holdases than wild-type. They exploit this observation to isolate complexes of sHsp and model substrate that are smaller (unusually) than the native assembly.

Overall, this is a nice study that provides a consistent picture of the importance of the oligomeric state of this chaperone and the transitions between these different states.

Some points to clarify:

1) The authors claim that: “The two positions of Loop 5/6 were noted in an earlier analysis of HSPB5-ACD structures and proposed to be correlated to differences in pH among structures (8). However, our analysis of the larger number of structures now available reveals that the Loop 5/6 position does not correlate with pH or with dimer register, suggesting it is a dynamic component of the structure under all conditions.”

This seems like a contradiction of their model that the position of this loop and its cross-interface interactions are important for determining the oligomeric state and that this changes with pH.

2) Does this loop show a higher mobility (sharper lines, longer T_2_ times) than the rest of the protein? This would indeed indicate a high flexibility.

3) What are potential hydrogen bonding partners of His 104 and how does this interaction stabilize the dimer and why is it disrupted by protonation. Since this is the central mechanistical result of this study, the authors should provide a possible model. A detailed figure of the structure around this loop region would also be important.

---

## [Author Response]

*1) The authors claim that: “The two positions of Loop 5/6 were noted in an earlier analysis of HSPB5-ACD structures and proposed to be correlated to differences in pH among structures (*[8]*). However, our analysis of the larger number of structures now available reveals that the Loop 5/6 position does not correlate with pH or with dimer register, suggesting it is a dynamic component of the structure under all conditions*.*”*

*This seems like a contradiction of their model that the position of this loop and its cross-interface interactions are important for determining the oligomeric state and that this changes with pH*.

We see how the statement as written could be construed as contradictory. When we went back to [8], we found they had not proposed the loop conformer to be a function of pH. We have therefore removed that inaccuracy and changed the text to: “Thus, among structures solved between pH 6.5 and 7.5, both loop conformations have been observed, suggesting that the loop is dynamic in the physiological pH range”.

*2) Does this loop show a higher mobility (sharper lines, longer T*_*2*_
*times) than the rest of the protein? This would indeed indicate a high flexibility*.

The ^15^N T_2_ values are, on average 30-40 msec for the ACD dimer. Only residues at the extreme termini show longer values. Importantly, resonances that arise from the dimer interface and from Loop 5/6 have *shorter* T_2_ values than the average (<30 msec), indicating that they undergo a change in their environment in the millisecond timescale. That is why we turned to the ^15^N relaxation-compensated CPMG relaxation dispersion measurements experiments, which specifically report on processes in the millisecond time regime. We have clarified this point in the subsection “NMR solution structure and ^15^N relaxation reveal conformational plasticity in HSPB5-ACD”.

*3) What are potential hydrogen bonding partners of His 104 and how does this interaction stabilize the dimer and why is it disrupted by protonation. Since this is the central mechanistical result of this study, the authors should provide a possible model. A detailed figure of the structure around this loop region would also be important*.

There are several backbone carbonyl groups (His-83, Glu-105, Glu-106) and several side chain groups (Arg-107 and Arg-116) in proximity to His-104 in members of the solution ensemble. We have added that information in the subsection “A conserved histidine is responsible for destabilization of the HSPB5-ACD dimer” and in the Discussion. The same group of interacting groups is also seen, in different combinations, in crystal structures that have the Loop 5/6-“up” conformation. We therefore believe that this is a dynamic network of interactions and are hesitant to show a detailed static picture that would only reflect one of the many possible sets of interactions as a figure in the main text. Nevertheless, analysis of structures in the Loop 5/6-“down” conformation reveals that potential interactors move away from His-104 in this state, namely the side chain of Arg-107, and others appear to move into proximity, namely the side chain of Glu-106. We propose that in both loop conformations His-104 sits at the center of a dynamic network of electrostatic and H-bonding interactions that is responsible for modulating the stability of the dimer interface. We have clarified this point in the Discussion and have added a supplemental figure, Figure 1–figure supplement 1 that shows the relevant region of ensemble members and an example of a loop-up and loop-down structure.